# Machine learning the metastable phase diagram of covalently bonded carbon

Srilok Srinivasan [1], Rohit Batra[1], Duan Luo [1], Troy Loeffler[1], Sukriti Manna[1,2], Henry Chan [1,2], Liuxiang Yang [3], Wenge Yang [3], Jianguo Wen [1✉], Pierre Darancet [1,4✉] & Subramanian K.R.S. Sankaranarayanan [1,2✉]

Conventional phase diagram generation involves experimentation to provide an initial estimate of the set of thermodynamically accessible phases and their boundaries, followed by use of phenomenological models to interpolate between the available experimental data points and extrapolate to experimentally inaccessible regions. Such an approach, combined with high throughput first-principles calculations and data-mining techniques, has led to exhaustive thermodynamic databases (e.g. compatible with the CALPHAD method), albeit focused on the reduced set of phases observed at distinct thermodynamic equilibria. In contrast, materials during their synthesis, operation, or processing, may not reach their thermodynamic equilibrium state but, instead, remain trapped in a local (metastable) free energy minimum, which may exhibit desirable properties. Here, we introduce an automated workflow that integrates first-principles physics and atomistic simulations with machine learning (ML), and high-performance computing to allow rapid exploration of the metastable phases to construct "metastable" phase diagrams for materials far-from-equilibrium. Using carbon as a prototypical system, we demonstrate automated metastable phase diagram construction to map hundreds of metastable states ranging from near equilibrium to far-from-equilibrium (400 meV/atom). We incorporate the free energy calculations into a neural-network-based learning of the equations of state that allows for efficient construction of metastable phase diagrams. We use the metastable phase diagram and identify domains of relative stability and synthesizability of metastable materials. High temperature high pressure experiments using a diamond anvil cell on graphite sample coupled with high-resolution transmission electron microscopy (HRTEM) confirm our metastable phase predictions. In particular, we identify the previously ambiguous structure of *n*-diamond as a cubic-analog of diaphite-like lonsdaelite phase.

[1] Center for Nanoscale Materials, Argonne National Laboratory, Lemont, IL 60439, USA. [2] Department of Mechanical and Industrial Engineering, University of Illinois, Chicago, IL 60607, USA. [3] Center for High Pressure Science and Technology Advanced Research, 100193 Beijing, P. R. China. [4] Northwestern Argonne Institute of Science and Engineering, Evanston, IL 60208, USA. ✉email: jwen@anl.gov; pdarancet@anl.gov; skrssank@uic.edu

Materials synthesis traditionally relies on "thermodynamic phase diagrams" to provide information about the stable phases as a function of various intensive state properties such as temperature, pressure, and chemical composition. The conventional method for generating a phase diagram involves experimentation to provide an initial estimate of phase boundaries followed by the use of phenomenological models to interpolate the available experimental data points and extrapolate to experimentally inaccessible regions. Such an approach combined with atomistic simulations and recent data-mining techniques has led to well-established exhaustive thermodynamic databases[1-3] for different materials—albeit limited to phases observed near thermodynamic equilibria. However, following synthesis and processing, or during operation, materials may be trapped in local minima of the energy landscape, that is, in metastable states (see Fig. 1a). Solid carbon is a prototypical system exhibiting such behavior, with numerous known metastable allotropes at room temperature and atmospheric pressure. Importantly, these allotropes have wide-ranging properties, from metals[4-7], semiconductors[8], topological insulators[9-11], to wide bandgap insulators[12]. Similarly, a vast and rich phase space of metastable structures for multi-component materials exists, some of these phases with potentially desirable properties, driving the need to go beyond near-equilibrium materials. Exhaustive "metastable phase diagrams", mapping the equation of states for phases without parent in thermodynamic equilibrium, are hence highly desirable.

Predicting, identifying, and mapping the free energy of metastable materials is a non-trivial and data-intensive task. The first challenge is to employ an efficient structure optimization algorithm capable of identifying both global (ground state) and local (metastable) minima of the energy landscapes in the configurational space. The next challenge is to map the free energy surface (i.e. the equation of state) for each of these metastable phases as a function of the intensive thermodynamic state variables ($P$, $T$ and $X$), over the range in which the phase information is desired. This step quickly becomes computationally prohibitive for large numbers of metastable configurations, and, in practice, requires a surrogate model, to approximate the free energy calculations of a more expensive first-principles-based approach (e.g. ab-initio molecular dynamics). After the equation of state for all the phases is computed, the final challenge is to classify and identify the phase boundaries and the domains of metastable equilibrium, i.e. the areas of the phase diagram in which a metastable structure is dynamically decoupled from lower energy structures.

Here, we report an automated framework that addresses the above challenges by integrating an evolutionary algorithm with first-principles calculations, machine learning (ML), and high-

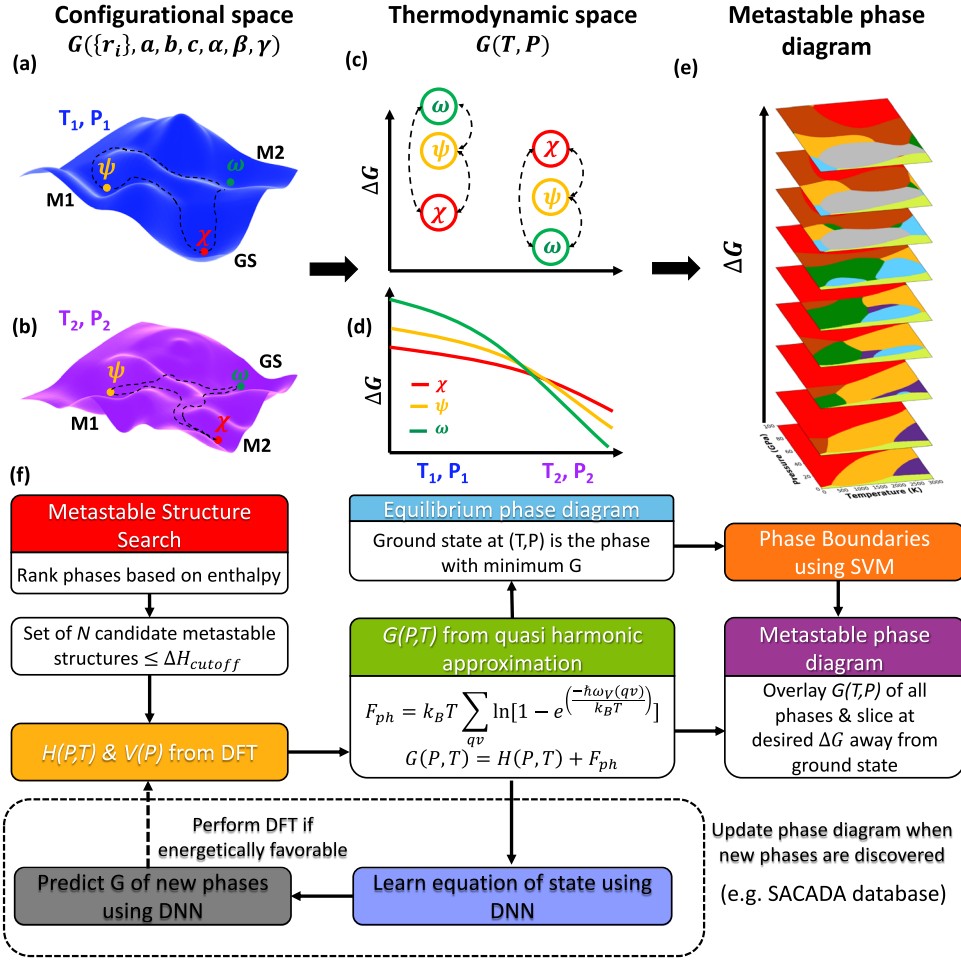

**Fig. 1 Our automated AI workflow for construction of metastable phase diagrams. a, b** Schematic illustration of the free energy landscape in the configurational space at different conditions ($T_1, P_1$), ($T_2, P_2$). $\{r_i\}$ refers to the coordinates of the basis atoms and $a, b, c, \alpha, \beta, \gamma$ are the lattice parameters. The phases corresponding to the minima are labeled $\chi, \psi, \omega$. GS, M1, and M2 stand for ground state, near-equilibrium and far-from equilibrium metastable phases. **c** Graph representation of the energy landscape. Nodes correspond to the phases and the edges contain the barrier height. **d** Equation of state for $\chi, \psi$ and $\omega$. **e** Illustration of the metastable phase diagram as a function of $\Delta G$. **f** Our workflow to identify metastable configurations and construct the metastable phase diagram. DFT, density functional theory; DNN, deep neural network; SVM, support vector machine.

performance computing to allow the exploration of the metastable materials and construct their phase diagrams. Our framework allows curation of metastable structures from published literature/databases and concurrently enables automatic discovery, identification, and exploration of the metastable phases of a material, and 'learns' their equations of state through a deep neural network. We apply our framework to the case of carbon –a system well-known to exhibit a large number of metastable allotropes–and map its metastable phase diagram in a large range of temperatures (0–3000 K), pressure (0–100 GPa) and excess free energy (up to 400 meV/atom above thermodynamic equilibrium). Importantly, we show that the proximal phases to thermodynamic equilibrium (within ~140 meV/atom) can be observed experimentally in high-pressure high temperature (HPHT) processing of graphite. In particular, we identify a new cubic-diaphite metastable configuration that explains the diffraction pattern of the previously reported $n$-diamond[13], demonstrating the potential of our approach to guide the synthesis of materials beyond equilibrium. We also demonstrate that our metastable phase diagrams can be used to identify both the relative stability of the various metastable phases as well as domains of synthesizability.

## Results

Our workflow is summarized in Fig. 1. We construct metastable phase diagrams with the chemical information of the periodic system as input, along with the range of pressure and temperature of interest.

As explained in detail below, we identify the metastable phases by sampling the configurational landscape at fixed thermodynamic conditions. The ground and metastable states at a given set of thermodynamic conditions $(T, P)$ correspond to global and local minima of the free energy in the configurational space, $G(\{r_i\}, a, b, c, \alpha, \beta, \gamma)$, where $a, b, c, \alpha, \beta, \gamma$ are the lattice parameters and $\{r_i\}$ are the position of the basis atoms. We then compute, at the identified minima, the Gibbs free energy in the thermodynamic space as a function of intensive variables $G(T, P)$; the free energy and the relative energetic ordering of its minima vary with $(T, P)$ as illustrated in Fig. 1a, b. Upon identification, the free energy and stability of these phases at $(T, P)$ is represented as a graph (Fig. 1c) with nodes corresponding to the free energy of the phases and the edges to the free energy barrier connecting them. This discrete thermodynamics representation is made continuous as a function of $(T, P)$, and the crossing points in the equation of states are automatically identified. Finally, we generate the full metastable phase diagram, $\mathcal{P}(T, P, \Delta G)$, where $\mathcal{P}$ is the most energetic phase within a free energy $\Delta G$ with respect to the ground state at a given $(T, P, \Delta G)$.

**Evolutionary structure prediction**. The first step in our workflow is to identify the periodic structures that are energetically favorable for a given chemical composition. We use an evolutionary search based on genetic algorithm—known to be efficient for periodic systems[14–18]. Briefly, evolutionary algorithms optimize the atomic arrangement $\{r_1, r_2, ... r_n\}$ and the lattice parameters $(a, b, c, \alpha, \beta, \gamma)$ of a population of structures over different regions of the energy landscape through genetic variations and selections over successive iterations. Hence, evolutionary algorithms are naturally suited to locate candidate metastable phases over the configurational space by evolving a pool of structures at the same time. Although $G$ includes both the temperature $(-TS)$ and pressure $(PV)$ contributions, for computational cost efficiency, we only include the effect of finite pressure in the selection of the offspring structures, by optimizing enthalpy at 0 K and fixed pressure, $H(T = 0\,K, P)$ —the entropic contributions being integrated at subsequent steps of our workflow.

We perform evolutionary structure search at several different pressures ($P = 0$ GPa, $P = 10$ GPa, and $P = 100$ GPa) independently by minimizing $H(T = 0\,K, P)$. More details regarding the evolutionary algorithm such as the genetic operations performed, selection criteria and other relevant parameters can be found in the supplementary methods (Section 1). All the distinct phases encountered during the search and their corresponding enthalpy values are recorded. Candidate metastable phases for further free energy calculations are identified from a collated list of structures from several independent evolutionary structure search at different pressures.

Diamond and graphite are the ground state phases observed in the experimental equilibrium phase diagram[19] of carbon. Apart from the equilibrium phases, our evolutionary search also identified metastable structures like the hexagonal diamond (lonsdaleite), several stacking combinations of cubic and hexagonal diamond (stacking disorder), distorted cubic diamond, distorted hexagonal diamond (diaphite), which are also observed in our HPHT experiments. In addition, we also identify Z-carbon[20,21], F-carbon[22,23], G-21[24], 10B[25], bct-carbon[26,27] and several other theoretically predicted phases of carbon (see Supplementary Table 1 for a complete list) within the Samara Carbon Allotrope Database (SACADA)[28,29]. At a given pressure, the structure with minimum enthalpy ($H_{\text{ground}}$) is the ground state at 0 K – in the case of carbon, graphite at 0 GPa. In this work, we focus on the bulk phases of carbon and exclude nanoscale clusters, such as fullerene C60, their ordered 3D configurations, amorphous or molten carbon phases, and metallic carbon at extreme pressures due to the prohibitively high computational cost required to estimate their free energies. At 0 K, we have $H(T = 0\,K, P) = G(T = 0\,K, P)$. Hence, we use cutoff criteria based on $H(T = 0\,K, P)$ to screen the candidate metastable phases for the subsequent free energy calculation. We define a $\Delta H_{\text{cut-off}}$ and only include structures whose enthalpy satisfies $H < H_{\text{ground}} + \Delta H_{\text{cut-off}}$ for the free energy calculation. In the present work, we set $\Delta H_{\text{cut-off}} = 670$ meV/atom, comparable to the excess enthalpy of C60 fullerene ($\Delta H_{\text{C60}} = 608$ meV/atom) that, we hypothesize, should be large enough to include the thermodynamically relevant metastable structures. Among the selected structures, we group geometrically similar and layered structures (for example hexagonal graphite, orthorhombic graphite, rhombohedral graphite) based on the radial distribution function, angular distribution function (see Supplementary Methods, Section 1.1.2) which further reduces the number of candidate structures for free energy calculation. After performing the above selection and grouping of structures, we narrow down to 505 candidate metastable structures for free energy calculations.

The candidate structures obtained from the evolutionary structure search are purely based on the enthalpy values at 0 K. However, the metastability of a structure at a finite temperature is determined based on the Gibbs free energy $G(T, P)$. The Gibbs free energy of each candidate screened from the previous step is computed across the temperature and pressure range of interest, and by including the temperature and entropic contributions of free energy to the enthalpy. The entropic part of the Gibbs free energy $(-TS(T, P))$ is obtained by modeling the atomic vibrations as a system of harmonic oscillators. The methodology for computing the vibrational free energy using first-principles density functional theory (DFT) can be found in the "Methods" section. We note that the harmonic model employed here neglects the anharmonic effects. The crystalline carbon phases considered here are expected to be weakly anharmonic to an extent that harmonic model is a good approximation[30]. For disordered and amorphous systems exhibiting non-negligible degree of anharmonicity, their free energy contribution should be included using an appropriate method[30].

**Phase-dependent equations of state through Deep Neural Networks**. As shown below, recently developed machine learning (ML) methods[31–34] for developing inter-atomic potentials[35–38] or estimating atomistic or molecular properties[39–41] can be used to compute Gibbs free energy as a continuous function of $T, P$. In particular, the equation of state of a phase can be predicted directly given only the 0 K structural information of a phase, allowing us to quickly estimate a $(T, P)$ region wherein a specific phase has low Gibbs free energy, and can be potentially realized in the experiments.

Deep neural networks (DNN) have been shown to show superior performance as compared to other regression techniques, particularly for problems that involve large volumes of data[31]. Thus, here, we develop a DNN model that takes as an input the smooth overlap of atomic positions (SOAP) representation[42] of a phase, along with $T$ and $P$ information. The DNN is trained on the Gibbs free energy data of 273 phases out of the 505 carbon phases. Regularization techniques, such as dropout and early stopping, were utilized to avoid overfitting. Some important low energy metastable phases, namely, $n$-diamond (S291), stacking fault diamond (S132) and 6B (S389), were intentionally left out from the training process and were used to evaluate the DNN performance. More details on the DNN architecture, training, and the SOAP descriptor are provided in the "Methods" section.

The phase diagrams constructed only considering the phases identified in this work are shown in Supplementary Fig. 14. However, using the DNN, we can rapidly screen the energetically relevant structures within SACADA database, not identified by our evolutionary search, and include them as well in the metastable phase diagrams. We perform explicit DFT calculation for those SACADA structures with sizes similar to the training set (less than 20 atoms/unitcell) and DNN predicted free energies less than 250 meV/atom with respect to cubic diamond. The metastable phase diagram, inclusive of the SACADA screened structures (18 phases), is shown in Fig. 2.

**Equilibrium phase diagram**. The phase diagrams are constructed by comparing the $G(T, P)$ of the candidate structures at a given $(T, P)$. The difference in free energies between phases corresponds to the energy separation of their respective minima in the configurational space ($\{r_i\}, a, b, c, \alpha, \beta, \gamma$) at a given $(T, P)$ (Fig. 1c, d). The final stage in our workflow is to clearly identify the phase boundaries as a function of $(T, P, \Delta G)$ separating the different phases. We use a multiclass SVM[43–47] (MSVM), using a non-homogeneous 3rd order polynomial kernel, which can classify multiple classes (phases) without relying on decomposition techniques (see Supplementary Methods, Section 1.2). The final equilibrium and metastable phase diagram with the decision boundaries drawn using MSVM are shown in Fig. 2.

We first validate our workflow by constructing equilibrium phase diagram and comparing against the experimental graphite-diamond phase boundary[19,48]. The color of the region in $(T, P)$ phase diagram corresponds to the color of the structures shown in Fig. 2a. As expected, from the experimental phase diagram[19,49], the cubic diamond phase is dominant at high pressure whereas graphite is more stable in the low-pressure region. Importantly, our predicted diamond-graphite phase boundary matches excellently with the experimental phase boundary[19] (dashed line in Fig. 2c).

The experimental phase diagram reported by Bundy[50], and later by others [19,49,51], describes two different crystallographic forms of diamond, namely the predominant "cubic" diamond alongside small fractions of "hexagonal" diamond. While cubic-diamond has a "ABCABC" stacking sequence of atomic layers,

hexagonal-diamond exhibits "ABAB" stacking. Although we compute the free energies individually, the different polytypes (4H, 6H, 15R, 21R, etc.) of hexagonal and cubic stacking combinations are collectively referred to as "stacking disorder" diamond in this work. We note that the stacking disorders in diamond are marginally stabilized ($\Delta G/k_B T < 0.2$) at high temperatures ($T > 1000$ K) and moderate to high pressures ($P > 50$ GPa) (see Supplementary Discussion, Section 2.1), where $k_B$ refers to the Boltzmann constant. The formation of mixtures of hexagonal and cubic diamond during high-pressure-high-temperature treatment of graphite has been reported by many others[52–62]. These observations are not surprising considering that the energetic differences between the stacking disorder and pure cubic diamond are only $0.2 \times k_B T$ or less. Such a small difference increases the likelihood (discussed below) of forming these phases at high temperatures.

**Metastable phase diagram**. We next construct the metastable phase diagram of carbon. We define the quantity $\Delta G_{GS_i}^{MS_j} = G_{MS_j} - G_{GS_i}$ as the difference in Gibbs free energy between a metastable structure $MS_j$ and the ground state $GS_i$ at given temperature and pressure, with $\Delta G_{GS_i}^{GS_i}(T, P) = 0$ and $\Delta G_{GS_i}^{MS_j}(T, P) > 0$ if $MS_j$ and $GS_i$ are distinct phases.

The probability of occurrence of a metastable phase at a given temperature $T$ is proportional to $\exp\left(-\frac{\Delta G_{GS_i}^{MS_j}}{k_B T}\right)$. We therefore construct a $\Delta G(T, P)$ surface, the projections of which can be used to derive the metastable phase diagram as a function of the degree of non-equilibrium from the corresponding equilibrium phase. We define a metastable phase diagram as the phase diagram obtained by projecting on $T - P$ plane, the phase $MS_j$ with closest $\Delta G_{GS_i}^{MS_j}(T, P)$ value compared to a given degree of non-equilibrium, $\Delta G$, and satisfies $\Delta G_{GS_i}^{MS_j}(T, P) < \Delta G$. In other words, by varying $\Delta G$, we are effectively taking slices of the overlaid free energy landscape (Fig. 1e) of all the structures. We do not exclude any phase during the construction of the metastable phase diagram and compare the free energies of all the structures. The metastable phase diagrams represent the most energetic phase accessible within that $\Delta G$. Experimentally, such phases can be accessed by using pulsed laser heating, in which the system undergoes phase transformation with the pulse providing the excitation energy to transition between local minima of the free energy (Fig. 1c).

The metastable phase diagram of carbon at $\Delta G$ equal to 40, 140, and 220 meV/atom (Fig. 2b–d) shows the appearance of metastable phases and their regions of metastability with respect to ground state. The stacking disorders in diamond are within $\Delta G = 40$ meV/atom with respect to pure cubic diamond. The lonsdaelite like hexagonal-diaphite phase and distorted cubic $n$-diamond, both of which also observed during high-pressure-high-temperature processing of graphite (see below), appear at a $\Delta G = 140$ meV/atom. At further higher $\Delta G = 220$ meV/atom, we observe several different metastable phases, which were also theoretically predicted, such as G92, G173, G178, G21, G120[24], W-carbon[63,64], H-carbon[65] and C2/m-16[66] phase. Beyond demonstrating the phase diversity, the metastable phase diagram allows us to determine an effective "low free-energy projection" of the phases likely to be kinetically stabilized for a set of experimental conditions (as shown in Section 3.3). Such representation of the metastability of different phases allows one to deduce the temperature-pressure ranges at which a phase is likely to be stabilized—and an estimate of minimum excitation

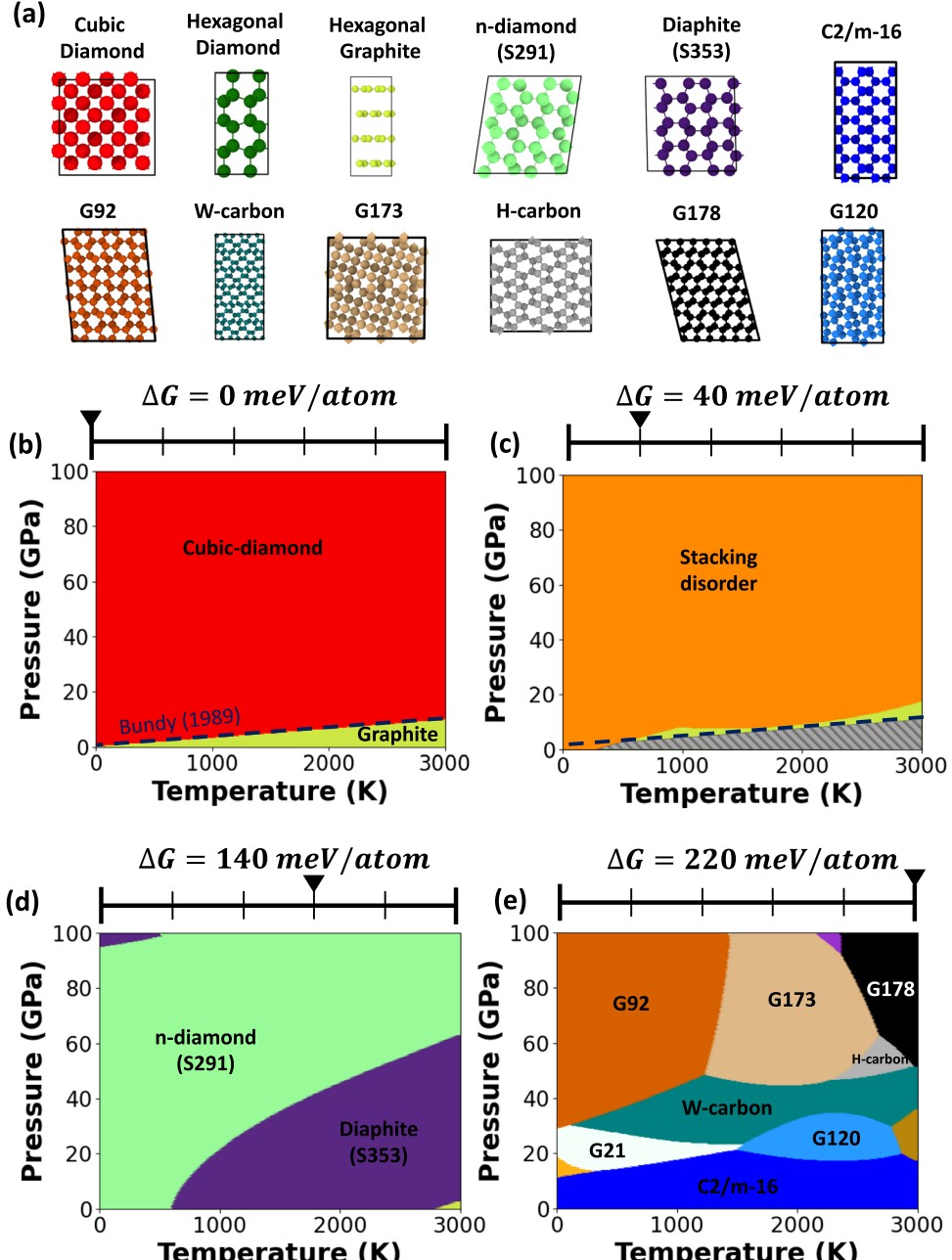

**Fig. 2 Machine learning a metastable phase diagram for carbon. a** Phases that appear within $\Delta G \leq 220$ meV/atom. **b** Equilibrium phase diagram with boundary fitted using MSVM. Equilibrium phase diagrams matches the experimental phase diagram[19,48]. **c–e** Metastable phase diagram (at a $\Delta G$ of 40, 140, and 220 meV/atom respectively) showing metastability of phases listed above. The color on the phase diagrams corresponds to the color of the structures shown in (**a**). Regions where no metastable phase is present other than the ground state are shaded in grey.

energies (from $\Delta G$) required to synthesize a metastable phase, thus guiding experiments at favorable conditions for synthesis.

We use the information derived from the metastable phase diagram to explain the experimental observations during laser heating induced phase transformation of hexagonal graphite in a pressurized diamond anvil cell[67,68]. As described in ref. [68] and ref. [67] the graphite crystal was heated to ≈1400 K by a YAG laser at the center of the crystal. Due to the Gaussian distribution of laser spot, a temperature gradient exists from the center to outside within a single laser spot in a given sample. In these recovered samples with the incomplete conversion of diamond, several metastable phases were identified by HRTEM as shown in Fig. 3. When pressurized, the graphite layers slide with respect to

each other to form orthorhombic and rhombohedral graphite (Fig. 3a)[67,69–71]. With further increase in temperature, the orthorhombic and rhombohedral graphite layers buckle to form interlayer bonds resulting in the formation of hexagonal or cubic diamond, respectively[69,70,72–78]. In practice, both the transformation pathways occur simultaneously, resulting in an intergrowth of cubic and hexagonal diamond[54,77,79–82], also known as the stacking disorder (shown in Fig. 3c).

As evidenced in ref. [68], the hexagonal diamond is actually a diaphite-like lonsdaelite phase with two different bond lengths[68]. One can again interpret this observation with the aid of our metastable phase diagram. Our structure model (S353) can explain experimental data in ref. [68] (see Supplementary

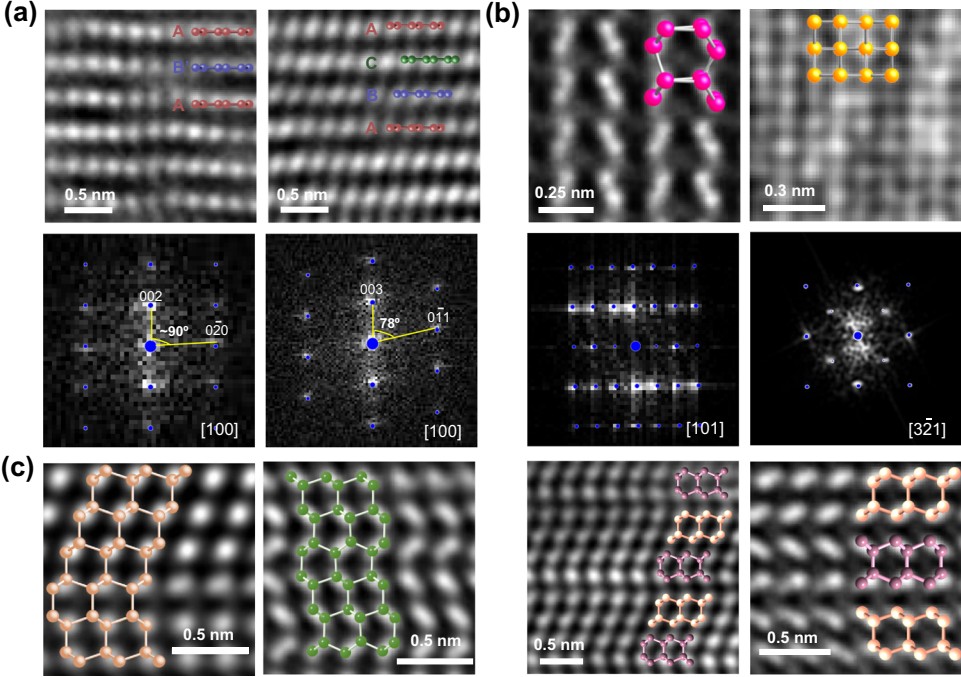

**Fig. 3 High-resolution TEM images of metastable phases of carbon. a** Orthorhombic-graphtie with AB′ stacking pattern and rhombohedral-graphite with ABC stacking along with the experimental and simulated diffraction patterns (blue circles)[67]. **b** Hexagonal-diaphite and cubic-Diaphite along with the experimental and simulated diffraction patterns(blue circles)[68]. **c** Different combinations of stacking patterns resulting from the simultaneous inter-growth of hexagonal and cubic-diamond[79,106].

Discussion, Section 2.5). The diaphite phase is easily accessible under the experimental conditions used (20 GPa, 1400 K) since it is metastable with $\Delta G$=140 meV/atom (a purple region in Fig. 2d) which is $\approx 0.86 \times k_B T$. We conjecture that graphite undergoes phase transformation, triggered by the excitation in experiments, into a accessible metastable phase which can be represented as excitation induced hopping from the global minima to a local minimum in the free energy landscape.

Furthermore, we observe a new cubic-diamond-like phase exhibiting the same diffraction pattern as the previously reported $n$-diamond[13]. New diamond ($n$-diamond) was proposed as a new carbon allotrope; its electron diffraction pattern matches that of cubic (Fd-3m) diamond apart from some additional reflections that are forbidden for diamond, indexed as {200}, {222} and {420}. The speculation of this new allotrope was first reported in 1991[13], but the exact crystal structure of $n$-diamond has remained a controversy despite several attempts to explain the $n$-diamond diffraction pattern[83–86]. Here, we attempt to explain the crystal structure of the metastable $n$-diamond using our metastable phase diagrams. Among all the phases that appear near the experimental conditions ($\approx$20 GPa, 1400K) in the metastable phase diagram at $\Delta G \approx 100$ meV/atom (Fig. 2e), the diffraction pattern of the S291 phase matches excellently with experiments (Fig. 3b). The S291 phase is a cubic analog of the diaphite-like lonsdaelite phase with two different bond lengths (Supplementary Discussion, Section 2.3). Similar to the diaphite-like lonsdaelite phase, cubic-diaphite is dynamically stable and has no imaginary phonon modes under a highly anisotropic pressure. Such anisotropic pressure can be explained by the buckling of basal planes, which induce the collapse of $c$-axis, equivalent to a huge increase in pressure in the out-of-plane direction. In fact, many rhombus voids within a single crystal, resulting from anisotropic pressure differentials, have been reported in diamond anvil experiments[67]. It is predicted that $n$-diamond nucleates at these

bent areas[87]. The observation of $n$-diamond phase as nanodomains with a size of ~100 nm suggests that they are not just defective cubic diamond and can potentially be stabilized as a standalone phase. Our experimental observation suggests that structure of n-diamond is a cubic diaphite with two different $sp^3$ bond lengths, which has not been reported before. This interpretation of the structure of cubic diaphite (n-diamond) can aid in synthesizing better quality n-diamond, as well as to understand the graphite-diamond transformation mechanism.

Hence, our framework not only correctly reproduces the dominant diamond and graphite phase in the equilibrium phase diagram, but also explains the observation of metastable phases in HPHT experiments. While one can do ad-hoc structure optimization to match experimental HRTEM images, the use of a metastable phase diagram not only accelerates phase identification by narrowing down the phase space but more importantly aids in discovering novel polymorphs. We note that while our framework does not guarantee an exhaustive search of all possible metastable structures, it allows for the inclusion of new metastable phases when encountered experimentally or theoretically.

Mapping the metastable phase diagram and inspecting the neighboring phases provides insight into possible phase transformation pathways and assists in selecting the appropriate starting material for targeted synthesis, thus accelerating computer-aided materials discovery.

## Discussion

**Domains of relative stability**. The metastable phase diagrams discussed above were generated by comparing the free energies of all the candidate phases. Often, materials scientists find it useful to consider only a select few phases of interest and inspect their relative probability of formation. For example, one may consider only two phases involved in a phase transition

## Relative stability of metastable phases

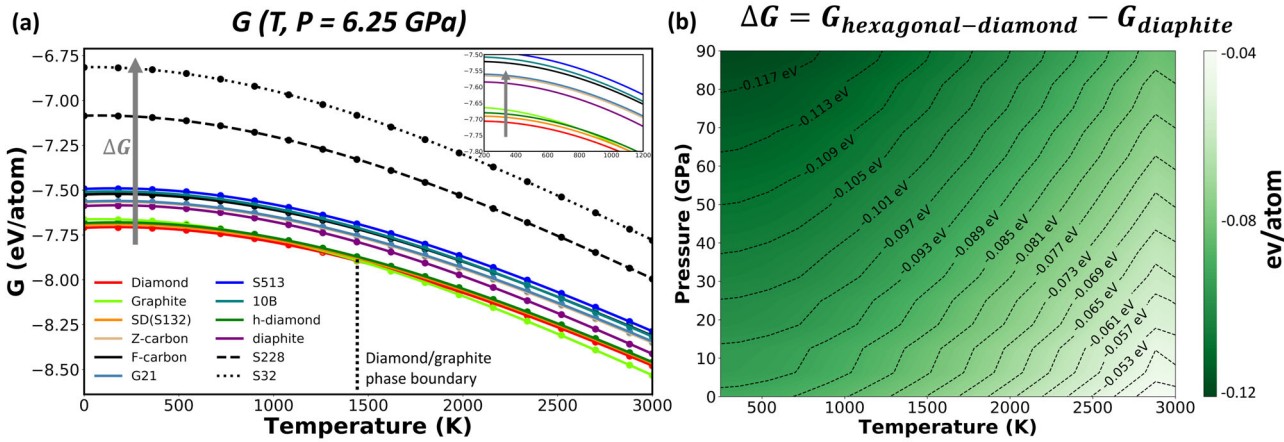

## Transformation barriers to metastable phases

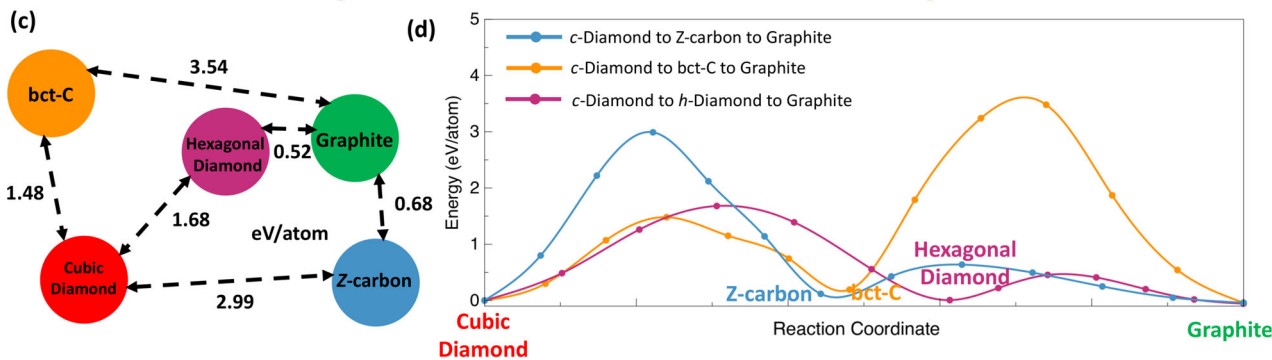

## Domains of synthesizability

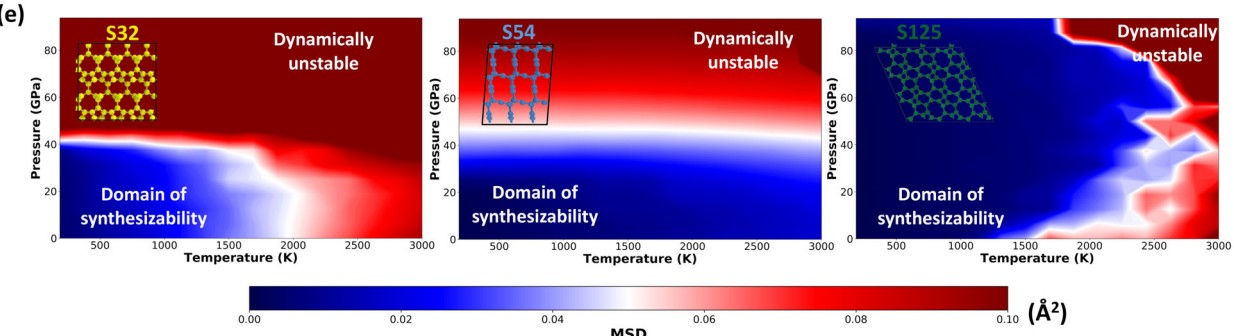

**Fig. 4 Information extracted from metastable phase diagram.** Relative stability of metastable phases: (**a**) $G(T, P = 6.25\,\text{GPa})$ of equilibrium and some representative metastable phases; (**b**) Relative stability between hexagonal diamond (green) and diaphite (purple) computed as $\Delta G = G_{\text{hex-diamond}} - G_{\text{diaphite}}$. **d** Transformation barriers to hexagonal diamond, bct-Carbon and Z-carbon metastable phases, starting from cubic-diamond and graphite, are used to construct a discrete (**c**) thermodynamic graph representing the transformation pathways and their respective barriers. **e** Domains of synthesizability based on dynamical stability for metastable phases S32, S54, and S125, respectively.

and study their relative stability, to estimate the phase transition line. The probability of observing a phase at a given pressure and temperature depends on its relative stability with respect to the competing phases. Figure 4a shows the free energy profile, at $P = 6.25$ GPa, of the near equilibrium metastable phases and some representative far from equilibrium metastable phases (S228, S32). The points where any two pair of lines intersect is the phase boundary between the corresponding phases. Free energies of distinct phases are separated by a finite $\Delta G$ (the degree of non-equilibrium). The

relative stability can also be considered as the projection, on the $T - P$ plane, of the distances between the free energy surfaces $G(T, P)$ for each phase. Figure 4b shows the map of the difference in the free energies $\Delta G = G_{\text{hex-diamond}} - G_{\text{diaphite}}$. Experimentally, diaphite is observed at moderate pressures and high temperatures whereas high-pressure conditions predominantly yield hexagonal diamond. Such information about the relative stability can aid in driving the synthesis process to yield a desired metastable phase, as opposed to a mixture of phases, by appropriately tuning the experimental conditions.

**Transformation barriers between metastable phases.** We note that our metastable phase diagrams, like any other phase diagram, contain information regarding the thermodynamics of the phases and only partial information regarding kinetics. In practice, the actual excitation required to overcome the kinetic barriers for phase transformation are larger than $\Delta G$. Once a list of competing metastable phases is identified over a $(T, P)$ of interest, an approximate energy barrier across any pair of phases can be estimated by matching the crystal structure to each other using the algorithm proposed by Stevanović et al.[88] and computing the energies of the images across the transformation pathway (see Supplementary Methods, Section 1.4). As an example, the transformation barriers for metastable hexagonal diamond, bct-C and Z-carbon phases, starting from ground state cubic-diamond and graphite, are shown in Fig. 4d. We construct a discrete thermodynamic graph representation (Fig. 4c) where the nodes represent individual phases and the length of the edges is proportional to the transformation barriers. Based on this graph, we find a lower transformation barrier for graphite to Z-carbon phase vs. cubic diamond to Z-carbon signifying graphite to be a preferred starting material for realizing the Z-carbon metastable phase. On the other hand, a cubic diamond would be relatively a better starting material to form bct-C. It is worth pointing out that metastable configurations of carbon have interesting electronic properties that range from metallic to semiconducting to insulating - which can be exploited if we can understand the thermodynamic landscape of such phases and the barriers to transform from one metastable phase (eg. metallic) to another (e.g. insulating). For example, graphite is electrically conducting whereas Z-carbon is insulating[65]. Similarly, diamond is insulating while bct-C carbon is predicted to be metallic[21,89]. Building such a network of metastable phases, related by their transformations pathways, can thus serve as a kinetics-based guide in determining the starting material during synthesis of a desired metastable phase. We also note that while the method proposed in Ref. [88] allows us to quickly estimate the approximate kinetic barriers by matching the crystal structures, a more accurate barrier height along with the minimum energy pathway and saddle point can be obtained from higher fidelity solid-state nudged elastic band (SSNEB)[90] calculations, once the competing transformations are identified above. As an example, we perform SSNEB calculation to compare graphite to hexagonal-diamond transformation with graphite to Z-carbon (Supplementary Methods, Section 1.5). The graphite to Z-carbon transformation has a larger kinetic barrier ($E_{\text{barrier}} = 0.47$ eV/atom) compared to graphite to hexagonal diamond transformation (eV/atom), in agreement with the findings using Stevanovic's method (0.68 eV/atom and 0.52 eV/atom, respectively). All the kinetic barriers computed here correspond to the concerted transformation mechanism.

**Domains of synthesizability.** The possibility of observing a phase at a given $T$ and $P$ depends on whether the crystal structure is retained or deformed due to melting or dynamical instability. In other words, the synthesizability is fundamentally limited by dynamical stability. We determine the dynamical stability of the metastable phases by inspecting the mean square deviation (MSD) of the atoms during MD simulations, performed over the temperature and pressure range of interest, using the long-range bond-order potential for carbon (LCBOP)[91]. The LCBOP potential reproduces the equilibrium phase diagram of carbon[48] and captures the equation of states of the phases (Supplementary Discussion, Section 2.2) considered in Fig. 4e. A metastable phase is considered dynamically unstable if the MSD is greater than 0.1 Å. In the context of Lindemann melting criteria[92,93], our choice of MSD cutoff corresponds to a Lindemann parameter of $\delta_L = 0.175$. Here, we define the domain

of synthesizability as the region in the $(T, P)$ space where a phase is dynamically stable. Figure 4e shows the domains of synthesizability of S32, S54, and S125. While the synthesizability of phases S32 and S54 is pressure limited, S125 is temperature limited. It should be noted that staying within the domain of synthesizability is a necessary, but not a sufficient condition for successful synthesis as there may be other factors limiting the synthesis. Similar upper limits for synthesizability, but based on the energetics of the amorphous phase, have been proposed in the past[94]. When a metastable phase is driven into a region of dynamical instability, it may transform into a neighboring metastable phase in the energy landscape or undergo melting to form an amorphous phase. Such theoretical bounds on the state variables $(T, P)$, where a phase is likely to be stabilized, are instructive for synthesizing a metastable phase of interest.

**Accelerating construction of metastable phase diagrams using machine learning.** The generation of metastable phase diagram relies on expensive free energy computations for a large number of competing phases. Inspired by the success of deep neural networks in achieving good performance on DFT datasets related to atomic energies and forces, we use a similar ML strategy in this work. Using ML-based surrogate models, we show that this process can be accelerated, and surrogate models that predict $G(T, P)$ can be constructed. Figure 5 presents the performance of the DNN model trained to predict $G(T, P)$ given only the structural information in the form of SOAP descriptor. The parity plots in Fig. 5a demonstrate the prediction accuracy (mean absolute error, MAE) achieved by the DNN model on the training as well as the test set. Notably, $n$-diamond (S291), S455 and 6B (S389) data were part of the test set and the good DNN performance for these cases illustrates its capability to capture the free energy surface of carbon. Further, in Fig. 5b we show that our DNN model is able to accurately predict the equation of state of phases in the test set, given only their structural information. The overall MAE across all phases in the test set was 37.1 meV/atom (Supplementary Discussion, Section 2.4) and was found to perform significantly superior to another baseline DNN model, which was fit to the coefficients of the free energy surface assuming its quadratic dependence with P and T. In many cases, high errors in the free energy predictions were observed at relatively higher pressures, as partially captured in Fig. 5. We note that learning Gibbs free energy as opposed to the potential energy or the enthalpy of a phase is fundamentally more challenging as it involves 2nd order derivatives with respect to energy. However, the encouraging performance of the DNN model on the test set indicates the overall promise of this approach. While the performance of DNN seems satisfactory, more rigorous work should be done in the future to find more suitable ML methods and the input structure fingerprints that improve the model performance further. Once such a surrogate model is trained, the free energy landscape of any new phase can be predicted orders of magnitude faster using only the structural information, thus, speeding up the process of constructing metastable phase diagrams.

In summary, we introduce an alternate representation of metastability by providing a free energy scale which helps identify both the metastable phase location and its extent of non-equilibrium. Such a representation is far more informative with regard to designing experiments and accelerating the discovery of metastable phases, which often display exotic properties. Our automated workflow allows for the construction of a "metastable phase diagram" by combining several synergistic computational approaches including a structural search based on genetic algorithms, deep learning accelerated high-throughput free energy calculations and multiclass support vector machines to classify phase boundaries. We demonstrate the efficacy of our computational approach by using a representative single-

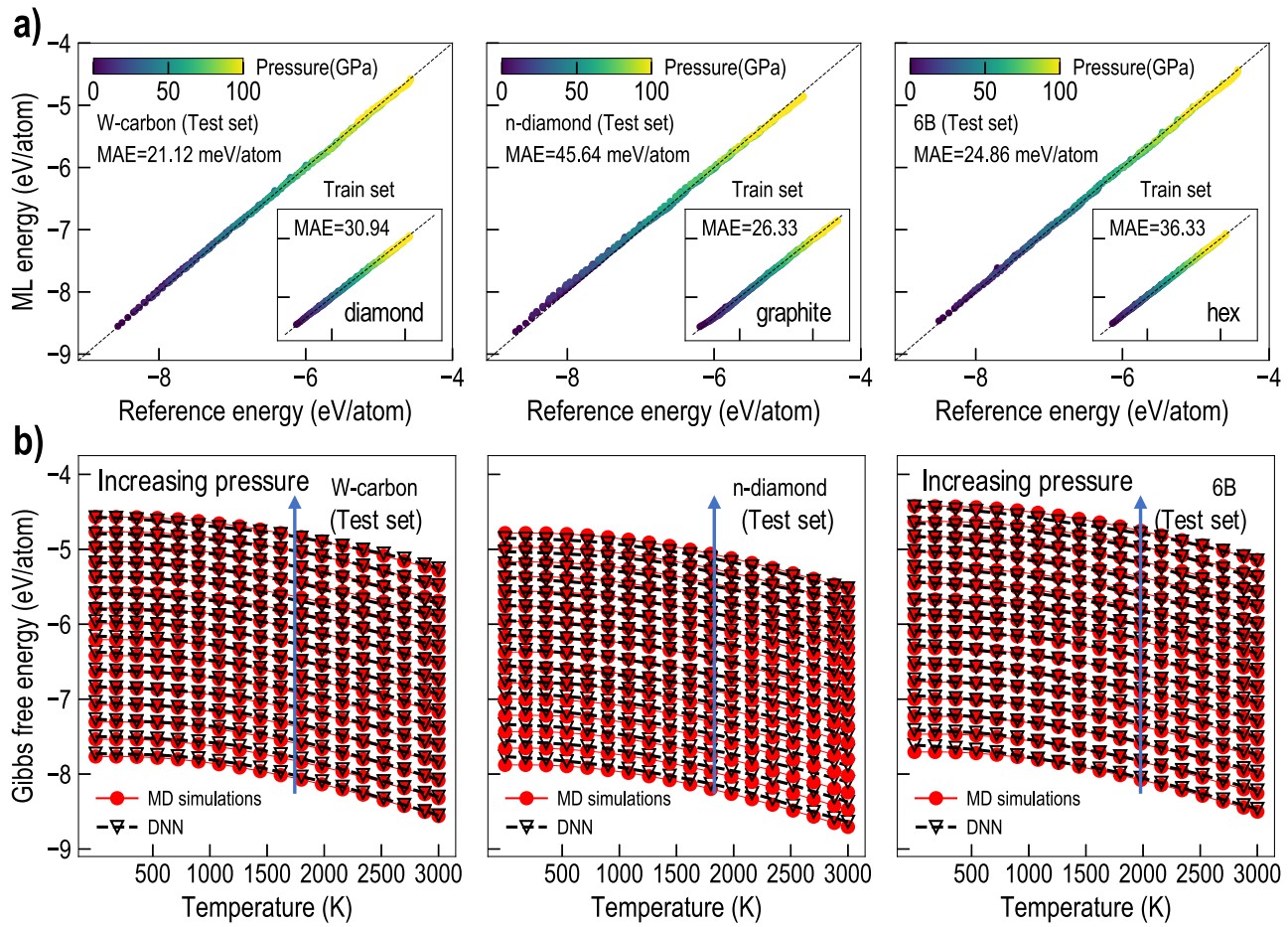

**Fig. 5 Performance of the DNN model to predict Gibbs free energy of different phases of carbon. a** Parity plot demonstrating prediction accuracy of DNN model against reference Gibbs energy dataset for different phases of C in the test or training set (inset). For better comparison, the range of x- and y-axes is kept consistent in the inset and the main panels. **b** Gibbs free energy predictions for the W-carbon, *n*-diamond (291) and 6B (389) phases for various temperature and pressures. Although these phases were part of the test set, DNN predicts their equation of states accurately. The blue arrows in panel (**b**) indicate the direction of increasing pressure. Abbreviation: MAE, mean absolute error.

component carbon system, whose equilibrium and metastable phases have been well studied in the past. We successfully predict the equilibrium phase diagrams, and using our metastable phase diagram, explain several experimental observations during high-pressure-high-temperature processing of graphite in diamond anvil cell. We propose a cubic-diaphitine structure, as a candidate phase to explain the diffraction pattern of *n*-diamond. In addition, we demonstrate that information about the relative stability of metastable phases and their synthesizability can be parsed from the metastable phase diagram. We also show that the phase diagram construction can be accelerated by orders of magnitude with the help of a surrogate ML model, which can reliably predict the equation of states, given only the structural information. Our framework lay the groundwork for computer-aided discovery and design of synthesizable metastable materials.

## Methods

**Evolutionary structure search**. Genetic algorithm (GA) was used to identify candidate structures, wherein an initial gene pool of crystal structures is randomly guessed and evolved in subsequent generations through genetic mutations or crossover between the fittest structures. Fitness of each organism in a given gene pool is evaluated as

$$f_i = \frac{H_i - H_{max}}{H_{min} - H_{max}} \qquad (1)$$

where $H_i$ is the enthalpy of the organism $i$, $H_{max}$ and $H_{min}$ are the maximum and minimum enthalpy in the current pool. The gene pool is ranked according to the fitness and parent structures are selected to undergo genetic variations to produce new offspring structures for the subsequent generation of gene pool. The selection probability

of each structure is based on the fitness:

$$p_i = \frac{f_i}{\sum_i f_i} \qquad (2)$$

The set of genetic operations used to build new generation of structures are: (i) Crossover variation, (ii) Structure mutation, and (iii) Number of atoms mutations. We perform the search with DFT as well as classical force fields (LCBOP) in independent GA runs. We collate a list of candidate structures from several independent evolutionary structure before computing the free energies. A detailed description of structure optimization procedure and genetic operations can be found in the Supplementary Methods.

**Free energy calculations**. The Gibbs free energy at a point $i$ in the thermodynamic phase space corresponding to a temperature ($T_i$) and pressure ($P_i$) can be written as

$$G_i(T_i, P_i) = H_i(T_i, P_i) - T_i S_i(T_i, V_i). \qquad (3)$$

where

$$H_i(T_i, P_i) = U_i + P_i V_i(T_i, P_i). \qquad (4)$$

Here we make the approximation

$$H_i(T_i, P_i) \approx H_i(T = 0K, P_i) \qquad (5)$$

In other words, we neglect the effects of thermal pressure and thermal expansion on enthalpy. This is a reasonable approximation considering that, for solids, the change in volume with respect to temperature and the associated $\Delta PV$ is $\ll U_i$. We can thus separate the temperature and pressure contribution to the Gibbs free energy as enthalpy and entropic contribution, respectively. Additionally, in solids with few atomic components, the vibrational contribution to the entropy

is the dominant one[95], and hence we make the approximation:

$$S(T, P) \approx S_{\text{vibrational}}(T, P), \quad (6)$$

If the atomic vibrations are modeled as harmonic oscillators, it follows that

$$F_{\text{Harmonic}} = U_{\text{Harmonic}} - TS_{\text{vibrational}} = \frac{1}{2}\sum_{qv}\hbar\omega(qv) + k_B T \sum_{qv}\ln\left[1 - \exp\left(-\frac{\hbar\omega(qv)}{k_B T}\right)\right] \quad (7)$$

The pressure range (0–100 GPa), over which the phase information is desired, is discretized into nine points—$P_0 \in$ {0.0 GPa, 13.3 GPa, 26.6 GPa, 40.0 GPa, 53.3 GPa, 66.6 GPa, 80.0 GPa, 93.3 GPa, 100.0 GPa}. For any given metastable phase, the equilibrium volume $V(P = P_0)$ and enthalpy $H(P = P_0)$ is obtained by relaxing the structure under and external pressure $P_0$ using density functional theory (DFT). The temperature-dependent entropic part of the free energy is obtained from Eq. (7) using the phonon modes computed at the equilibrium volume $V(P = P_0)$.

The total Gibbs free energy $G(T = T_0, P = P_0)$ is obtained by summing the enthalpy $H(P = P_0)$ and the vibrational free energy $F_{\text{Harmonic}}$.

The DFT calculations were performed using the VASP package[96] under Perdew–Burke–Ernzerhof[97] approximation with optB86b-vdW[98,99] exchange functional to include the van der Waals interactions. All calculations were done with an energy cutoff of 600 eV. A dense $K$-point grid defined by $n_{\text{atoms}} \times n_{\text{kpoints}} \approx 6000$, where $n_{\text{atoms}}$ is the number of atoms in the primitive cell and $n_{\text{kpoints}}$ is the number of $k$-points, is employed[100,101]. The phonon modes were computed from the Hessian matrix, obtained from density functional perturbation theory, using the PHONOPY package[102].

All the calculations were performed petascale supercomputer, *Theta*, at Argonne Leadership Computing Facility (ALCF). While performing high throughput DFT calculations can quickly become expensive, when such computing resources are not readily available, one can use cheap models like semi-empirical[91] or machine learnt classical force fields[33,42] which offers a reasonable compromise between computational cost and accuracy.

**Deep neural network**. A deep neural network (DNN) was used to learn the Gibbs free energy of different phases of carbon. It consisted of 8 fully connected (dense) hidden layers with 128, 256, 512, 512, 512, 512, 64, and 32 neurons, respectively, as shown in Supplementary Fig. 3. Relu activation function was used in all layers, except the input and the output layers. Many dense layers were followed by batch normalization to assist the training of such a large network. The input layer consisted of smooth overlap of atomic positions[42] (SOAP) of the 0 K and 0 GPa structure of a phase, and the normalized $T$ and $P$ value. The SOAP fingerprint was obtained using the python library Dscribe[103] with the following parameter settings: $r_{\text{cut}} = 6\text{Å}$, $n_{\text{max}} = 6$ and $l_{\text{max}} = 4$, where $r_{\text{cut}}$ is the cut-off radius for the atomic neighborhood around the concerned atom, $n_{\text{max}}$ is the number of radial basis functions (spherical Gaussian type orbitals) and $l_{\text{max}}$ is the maximum degree of spherical harmonics. This resulted in a SOAP fingerprint vector for each atom, which was averaged using the "inner" averaging scheme (average over atomic sites before summing up the magnetic quantum numbers) to obtain a 105-dimensional configuration fingerprint for each phase. To account for the large variation across the different features of the fingerprint, each feature was normalized by removing the mean and scaling to unit variance, as obtained from the data in the training set. $T$ and $P$ values were included as two additional features, overall resulting in a 107-dimensional input fingerprint to the DNN.

The output layer consisted of a single neuron describing the DNN predicted Gibbs free energy of a phase at the input $T$ and $P$ values. The DNN was trained using Adam optimization algorithm[104] with the mean absolute error chosen as the loss function definition. Free energy data corresponding to 273 phases was used to train the model, while that for 19 and 31 phases was used as the validation and test set, respectively. A few important phases, such as diamond, graphite and diaphite (S353), were part of the training set, while others, including $n$-diamond (S291), stacking-disorder (S132), and 6B (S389), were part of the test set. Since some phases were found to be dynamically unstable at different P and T conditions, caution was taken to only include free energy training data when the phase was stable. The number of training epochs was determined by monitoring the model performance on the validation set and multiple dropout layers (with values of 0.1-0.3) were used for regularization purposes. The DNN code was implemented in Tensorflow[105]. The overall performance of the DNN model on the training as well as the test set is presented in Supplementary Fig. 3.

## Data availability

All data supporting the findings of this study are available within the Supplementary Information. Any further related information can be provided by the authors upon reasonable request.

## Code availability

The code used to train DNN model are provided in the GitHub repository: https://github.com/Srilok/Machine-learning-Metastable-Phase-Diagram. The scripts and framework to construct the metastable phase diagram are available from the authors upon reasonable request.

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

## Acknowledgements

This material is based on work supported by the DOE, Office of Science, BES Data, Artificial Intelligence and Machine Learning at DOE Scientific User Facilities program (Digital Twins and Collaborative MLExchange). Use of the Center for Nanoscale Materials, an Office of Science user facility, was supported by the U. S. Department of Energy, Office of Science, Office of Basic Energy Sciences, under Contract No. DE-AC02-06CH11357. This research also used resources of the Argonne Leadership Computing Facility at Argonne National Laboratory, which is supported by the Office of Science of the U.S. Department of Energy under contract DE-AC02-06CH11357. The authors (SKRS and TDL) would like to acknowledge the Air Force Office of Scientific Research (AFOSR) for funding this research under Award FA9550-20-1-0332, with Dr. Chipping Li as the program manager. This research used resources of the National Energy Research Scientific Computing Center, a DOE Office of Science User Facility supported by the Office of Science of the U.S. Department of Energy under Contract No. DE-AC02-05CH11231. This material is based upon work supported by Laboratory Directed Research and Development (LDRD) funding from Argonne National Laboratory, provided by the Director, Office of Science, of the U.S. Department of Energy under Contract No. DE-AC02-06CH11357. We gratefully acknowledge the computing resources provided on Fusion and Blues, high performance computing clusters operated by the Laboratory Computing Resource Center (LCRC) at Argonne National Laboratory.

## Author contributions

SS, RB, TL, PD, SKRS designed and conceived the project. SS performed the structure identification, free energy calculation (with input from SM) and metastable phase diagram construction. HC performed the grouping of structures. RB developed the deep learning model to learn the free energies and equation of state for metastable structures. SM prepared the training data for the deep learning model. DL, LY, WY and JGW performed the high pressure and high temperature experiments and characterization. All the authors contributed to the data analysis and preparation of the manuscript. SS, RB, PD, JGW and SKRS wrote the manuscript with input from other co-authors. SKRS and PD supervised the overall project.

## Competing interests

The authors declare no competing interests.
