## [Peer Review File · Nature Communications]

REVIEWERS' COMMENTS

Reviewer #1 (Remarks to the Author):

I appreciate that the authors have carefully answered all my concerns in their revision and reply letter. After the revision, I believe the authors have greatly improved the presentation regarding the computational methodologies and impacts on carbon research. That said, I would like to reserve my opinion on the use of DL approach. According to the results as shown in their reply letter, it seems that DL does not significantly improve the accuracy with respect the relative simple GPR model. If the authors is concerned about the large volume of data, a shallow NN with properly engineered descriptor (not deep NN) may behave equally better. In fact, I am also a bit worried that the improvement from DL is due to overfitting. At this stage, I won't suggest the authors to try any other new approach since they have already spent enormous time on it. However, the author would be better neutral about the use of DL. Don't leave the readers an impression that DL is a must-go approach and mention that there may exist alternative approaches with simpler framework.

From computational point of view, my judgement is that the paper is well written and can be considered for a publication in Nature Comm.

Reviewer #5 (Remarks to the Author):

The paper introduces the idea of a metastable "phase diagram" and charts out the possible "phases" which might exist there. They are characterised by differences in free energy from the stable state at a given P,T condition. This is a very valuable and worthwhile thing to try to do.

Temperature dependence is determined by the harmonic phonon method, which appears to preclude a very wide range of high-T phases - anything which is dynamically stabilised, anything involving molecular rotations, melting, and (probably) amorphous materials etc. They apply the

method to carbon, excluding the fullerene phases ostensibly by cutoff in ΔG . They hypothesize, that this "should be large enough to include the thermodynamically relevant metastable structures". This is a bizarre claim given the known existence of fullerene crystals. They also exclude fullerenes also by not searching for large enough unit cells, because they test DFT only up to 20 atoms per cell.

I appreciate that large scale structure searches are expensive, and fullerenes may be troublesome for their method - but why then choose carbon in the first place if you know you won't scan the whole range of possibilities?

They cite as a reason for choosing carbon that "Importantly, these allotropes have wide-ranging properties, from metals [4–7], semiconductors [8], topological insulators [9–11], to wide band gap insulators [12]". It's an excellent reason. Unfortunately, after this sentence in the introduction no further mention of metals, semiconductors or topological insulators is made.

The fact that the method is so limited in scope is not necessarily a block on publication, but it does mean the title "Machine Learning the Metastable Phase Diagram of Materials" misleadingly over claims what they have done. "Machine Learning the Metastable Phase Diagram of Covalently bonded Carbon" would be more accurate and honest.

I don't see amorphous carbon on the phase diagram. There is a sentence

"For disordered and amorphous systems exhibiting non-negligible degree of anharmonicity, their free energy contribution should be included using an appropriate method [33]". But there's no further mention of ref [33] (a review) or an "appropriate method". So while I agree that it should be done, I strongly suspect that they haven't done it.

They explain, correctly, that G depends on both T and P . But when they actually apply the workflow: "for computational cost efficiency, we only include the effect of finite pressure in the selection". And no discussion of how this could be included in the paper.

Overall, there is plenty of interesting work here relevant to the low temperature- high pressure phases of carbon. The method may be applicable more generally, but since they chose carbon, and then molten, amorphous, fullerene and metallic carbon at extreme pressures are all omitted, this is not convincingly demonstrated. I think it could be published in Nature Communications if the excessive claims of generality of the method were removed.

Responses to Referees' Report for Machine Learning the Metastable Phase Diagram of Materials

April 15, 2022

Reviewer #1:

I appreciate that the authors have carefully answered all my concerns in their revision and reply letter. After the revision, I believe the authors have greatly improved the presentation regarding the computational methodologies and impacts on carbon research. That said, I would like to reserve my opinion on the use of DL approach. According to the results as shown in their reply letter, it seems that DL does not significantly improve the accuracy with respect the relative simple GPR model. If the authors is concerned about the large volume of data, a shallow NN with properly engineered descriptor (not deep NN) may behave equally better. In fact, I am also a bit worried that the improvement from DL is due to overfitting. At this stage, I won't suggest the authors to try any other new approach since they have already spent enormous time on it. However, the author would be better neutral about the use of DL. Don't leave the readers an impression that DL is a must-go approach and mention that there may exist alternative approaches with simpler framework. From computational point of view, my judgement is that the paper is well written and can be considered for a publication in Nature Comm.

Author reply: We agree with reviewers' opinion that more rigorous comparison between different ML strategies and input structure fingerprint is required to confidently find the best ML approach for this learning problem. Accordingly we have modified the main manuscript. Moreover, we believe that the presented NN models do not have an overfitting issue because of the following two reasons: 1) we adopted several established techniques to avoid over-fitting such as dropout, validation set, etc. 2) the good predictions achieved on the SACADA dataset (which was not part of the NN model training exercise) is indicative of the high quality of the models trained. This would not be the case if the NN models were overfit.

Changes: We have added the following sentences in the revised manuscript:

- In Discussion, line 346, *"Inspired by the success of deep neural networks in achieving good performance on DFT datasets related to atomic energies and forces, we use a similar ML strategy in this work."*
- In Discussion, line 362, *"While the performance of DNN seems satisfactory, more rigorous work should be done in the future to find more suitable ML methods and the input structure fingerprints that improves the model performance further."*

Reviewer #5:

The paper introduces the idea of a metastable "phase diagram" and charts out the possible "phases" which might exist there. They are characterised by differences in free energy from the stable state at a given P,T condition. This is a very valuable and worthwhile thing to try to do. Temperature dependence is determined by the harmonic phonon method, which appears to preclude a very wide range of high-T phases - anything which is dynamically stabilised, anything involving molecular rotations, melting, and (probably) amorphous materials etc. They apply the method to carbon, excluding the fullerene phases ostensibly by cutoff in ΔG . They hypothesize, that this "should be large enough to include the thermodynamically relevant metastable structures". This is a bizarre claim given the known existence of fullerene crystals. They also exclude fullerenes also by not searching for large enough unit cells, because they test DFT only up to 20 atoms per cell.

Author reply: In this work, we have mainly considered bulk carbon phases and excluded nanoscale metastable structures such as fullerenes (C_{60}) or their ordered 3D configurations. These were mainly due to the computational cost associated with much larger simulation cells needed to model such metastable structures within the density functional theory (DFT) framework. Also, considering the computational expense of free energy calculations using first principles DFT, it was necessary to choose an enthalpy cutoff to keep the number of calculations tractable. Moreover, the computational complexity of the DFT calculation also increases with the size of the unit cell making the calculations further difficult to perform in a high throughput settings. While the phase with large unit cells (e.g. 3D fullerene configurations) are not considered in our calculations, we believe the main contribution of this work is conceptualize a "metastable phase diagram" as a tool to accelerate discovery of novel materials with desired properties

Changes: We have added the following sentence to clarify this

- In Results, line 120, *"In this work, we focus on the bulk phases of carbon and exclude nanoscale clusters, such as fullerene C_{60} , their ordered 3D configurations, amorphous or molten carbon phases, and metallic carbon at extreme pressures due to the prohibitively high computational cost required to estimate their free energies."*

I appreciate that large scale structure searches are expensive, and fullerenes may be troublesome for their method - but why then choose carbon in the first place if you know you won't scan the whole range of possibilities?

They cite as a reason for choosing carbon that "Importantly, these allotropes have wide-ranging properties, from metals [4–7], semiconductors [8], topological insulators [9–11], to wide band gap insulators [12]". It's an excellent reason. Unfortunately, after this sentence in the introduction no further mention of metals, semiconductors or topological insulators is made.

Author reply: Carbon has a multitude of metastable structures with interesting properties and hence was chosen for this study. For reasons stated in the previous comment, we excluded 3D fullerene arrangements but consider ~ 1000 other metastable configurations. In principle, one can never be certain if we have scanned all possible metastable structures and they will continue to be discovered both experimentally and theoretically. Our framework, however, allows for inclusion of such metastable configurations. In future, as computations become more efficient, we should be able to include fullerenes as well and identify their phase space (T,P) in the metastable phase diagram of carbon. We have added a few sentences in the manuscript to address this. Regarding the second statement in this comment, we have added more sentences in the revised manuscript to clarify that the metastable configurations of carbon have interesting electronic properties that range from metallic to semiconducting to insulating - which can be exploited if we can understand the thermodynamic landscape of such phases and the barriers to transform from one metastable phase (eg. metallic) to another (e.g. insulating).

Changes: We have added the following sentences

- In Results, line 271, *"We note that while our framework does not guarantee an exhaustive search of all possible metastable structures, it allows for the inclusion of a new metastable phases when encountered experimentally or theoretically."*
- In Discussion, line 308, *"It is worth pointing out that metastable configurations of carbon have interesting electronic properties that range from metallic to semiconducting to insulating - which can be exploited if we can understand the thermodynamic landscape of such phases and the barriers to transform from one metastable phase (eg. metallic) to another (e.g. insulating). For example, graphite is electrically conducting whereas Z-carbon is insulating [1]. Similarly, diamond is insulating while bct-C carbon is predicted to be metallic [2, 3]."*

The fact that the method is so limited in scope is not necessarily a block on publication, but it does mean the title "Machine Learning the Metastable Phase Diagram of Materials" misleadingly over claims what they have done. "Machine Learning the Metastable Phase Diagram of Covalently bonded Carbon" would be more accurate and honest.

Author reply: We thank the reviewer for their suggestion. We have now changed the title of the manuscript

Changes: The title has been changed to "Machine Learning the Metastable Phase Diagram of Covalently bonded Carbon"

I don't see amorphous carbon on the phase diagram. There is a sentence "For disordered and amorphous systems exhibiting non-negligible degree of anharmonicity, their free energy contribution should be included using an appropriate method [33]". But there's no further mention of ref [33] (a review) or an "appropriate method". So while I agree that it should be done, I strongly suspect that they haven't done it.

Author reply: Since the free energies were computed from a periodic model, to reliably calculate free energies for amorphous phase, we require (i) a large unit cell to ensure there are no short range order and (b) sampling over several different configurations. Both of these factors make the free energy calculation of amorphous system using DFT prohibitively expensive. So we have not performed those calculations here and would be considered in future work to account for crystalline (metastable) to amorphous transformations

They explain, correctly, that G depends on both T and P . But when they actually apply the workflow: "for computational cost efficiency, we only include the effect of finite pressure in the selection". And no discussion of how this could be included in the paper.

Author reply: The above statement is true for the selection of candidate phases from evolutionary structure search since we are minimizing enthalpy. We would like to clarify that both the temperature effects and the entropic contributions are accounted for in the harmonic model for Gibbs free energy. Also, pressure effects are considered in the free energy calculations since we explicitly consider the pressure induced volume and configurational change in our free energy calculations. We have clarified this in the revised manuscript.

Changes: We have added the following sentence in the revised manuscript

- In Results, line 136, "*The Gibbs free energy of each candidate screened from the previous step is computed across the temperature and pressure range of interest, and by including the temperature and entropic contributions of free energy to the enthalpy*"

Overall, there is plenty of interesting work here relevant to the low temperature- high pressure phases of carbon. The method may be applicable more generally, but since they chose carbon, and then molten, amorphous, fullerene and metallic carbon at extreme pressures are all omitted, this is not convincingly demonstrated. I think it could be published in Nature Communications if the excessive claims of generality of the method were removed

Author reply: We thank the reviewer for their nice words about our work and for recommending publication in Nature Communications. We agree with the reviewer and have changed the title as per the reviewer suggestion and also explicitly stated in the revised manuscript that molten, amorphous, fullerene and metallic carbon at extreme pressures are not considered in our current metastable phase diagram workflow.

Changes: As pointed out by the reviewer, due to the computational challenges that remains to be addressed, we removed the following statement from the abstract and conclusion respectively

- The workflow presented here is general and broadly applicable to single and multi-component systems.
- Our data-driven approach is fairly general and applicable to other chemical systems with negligible configurational entropy and weak anharmonicity

References

- [1] C. He, L. Sun, C. Zhang, X. Peng, K. Zhang, and J. Zhong, "New superhard carbon phases between graphite and diamond," *Solid State Communications*, vol. 152, no. 16, pp. 1560–1563, 2012. [Online]. Available: <https://www.sciencedirect.com/science/article/pii/S0038109812003432>
- [2] C. He, L. Sun, C. Zhang, and J. Zhong, "Two viable three-dimensional carbon semiconductors with an entirely sp² configuration," *Phys. Chem. Chem. Phys.*, vol. 15, pp. 680–684, 2013. [Online]. Available: <http://dx.doi.org/10.1039/C2CP43221H>
- [3] Z. Zhao, B. Xu, X.-F. Zhou, L.-M. Wang, B. Wen, J. He, Z. Liu, H.-T. Wang, and Y. Tian, "Novel superhard carbon: C-centered orthorhombic c₈," *Phys. Rev. Lett.*, vol. 107, p. 215502, Nov 2011. [Online]. Available: <https://link.aps.org/doi/10.1103/PhysRevLett.107.215502>